# Cortical and Subcortical Brain Volumes Partially Mediate the Association between Dietary Composition and Behavioral Disinhibition: A UK Biobank Study

**DOI:** 10.3390/nu13103542

**Published:** 2021-10-09

**Authors:** Daan van Rooij, Lizanne Schweren, Huiqing Shi, Catharina A Hartman, Jan K Buitelaar

**Affiliations:** 1Donders Center for Brain, Cognition and Behaviour, Department of Cognitive Neuroscience, RadboudUMC, 6525 HB Nijmegen, The Netherlands; Huiqing.Shi@radboudumc.nl (H.S.); jan.buitelaar@radboudumc.nl (J.K.B.); 2Interdisciplinary Center Psychopathology and Emotion Regulation, University Medical Center Groningen, 9700 RB Groningen, The Netherlands; l.j.s.schweren@umcg.nl (L.S.); c.a.hartman@umcg.nl (C.A.H.)

**Keywords:** behavioral disinhibition, diet, structural, MRI, UKbiobank

## Abstract

Behavioral disinhibition is observed to be an important characteristic of many neurodevelopmental and psychiatric disorders. Recent studies have linked dietary quality to levels of behavioral inhibition. However, it is currently unclear whether brain factors might mediate this. The current study investigates whether cortical and subcortical brain volumes mediate part of the association between dietary composition and behavioral disinhibition. A total of 15,258 subjects from the UK Biobank project were included in the current study. Dietary composition and behavioral disinhibition were based on Principle Component Analyses of self-reported dietary composition). As a further data reduction step, cortical and subcortical volume segmentations were input into an Independent Component Analysis. The resulting four components were used as mediator variables in the main mediation analyses, where behavioral disinhibition served as the outcome variable and dietary components as predictors. Our results show: (1) significant associations between all dietary components and brain volume components; (2) brain volumes are associated with behavioral disinhibition; (3) the mediation models show that part of the variance in behavioral disinhibition explained by dietary components (for healthy diet, restricted diet, and high-fat dairy diet) is mediated through the frontal-temporal/parietal brain volume component. These results are in part confirming our hypotheses and offer a first insight into the underlying mechanisms linking dietary composition, frontal-parietal brain volume, and behavioral disinhibition in the general adult population.

## 1. Introduction

Behavioral disinhibition refers to the problematic and uncontrolled expression of impulsive behavior [1] and is observed to be an important characteristic of several neurodevelopmental and psychiatric disorders. Disorders like Attention-Deficit Hyperactivity Disorder (ADHD), Obsessive-Compulsive Disorder (OCD), Tourette’s Disorder, Mania and Substance Use Disorder (SUD) are characterized by a poor ability to control behavior [2,3,4]. Severe behavioral disinhibition is also more generally linked to increased negative health outcomes and increased mortality over the lifespan [5,6], demonstrating the need to study factors underlying behavioral disinhibition at the population level. 

A wealth of studies have indicated that lifestyle factors, including dietary composition, may be associated with cognitive performance and mental health factors. Specifically, previous work has largely focused on the effects of a healthy diet and physical fitness on improved cognitive performance and reduced symptoms of major depressive disorder [7,8,9] and anxiety disorders [10,11]. More recently, research has shown neurodevelopmental disorders like ADHD [12,13] or Tourette Disorder [14,15], and impulse control disorders like substance use disorder [16,17] to be also influenced by dietary composition, indicating an effect of diet on a broad range of mental health disorders. We postulate here that behavioral disinhibition may play a key role as an overarching factor linking the association between diet and mental health. Indeed, studies have consistently linked overall dietary quality to levels of behavioral inhibition [18,19,20,21]. In particular, a recent large-scale population-based study using the UKbiobank cohort was able to show associations between behavioral disinhibition and four main dietary composition components, namely overall diet quality, presence of specific dietary restrictions, meat/fish intake, and high-fat dairy intake [22]. 

A second factor to take into consideration when investigating the link between mental health and diet is the brain. Several studies have established associations between a poor diet and lower brain volumes [23,24,25]. Studies using a Mediterranean diet intervention report larger prefrontal brain regions, crucially implicated in behavioral disinhibition [26], in cross-sectional cohorts [27,28]. Therefore in the current study, we aim to investigate the neurobiological mechanisms underlying the association between diet and behavioral disinhibition by testing the links between diet, brain morphology, and behavioral disinhibition. An extensive body of literature shows that behavioral disinhibition is associated with both structural and functional changes in the brain [29,30,31,32,33,34,35,36]. Specifically, differences in the cortical-striatal pathways associated with cognitive control have been associated with high levels of disinhibited behavior. Both subcortical volumes of the striatum and frontal cortical thickness have been found to be decreased in association with disinhibition [37,38]. Functionally, frontal-parietal and striatal hypoactivation has been found in impulsive subjects during inhibition tasks [31,39,40,41,42,43]. These studies suggest that both diet and disinhibition are associated with the frontal–striatal pathways in the brain, making this a likely target to mediate the association between diet and disinhibition. 

The current study, therefore, aims to test whether the association between dietary composition and behavioral disinhibition is mediated by cortical and subcortical brain volumes in a large scale adult population sample (UKbiobank [44]). We hypothesize that (1) dietary composition is associated with brain volumes. (2) Brain volumes and (3) dietary composition are each associated with behavioral disinhibition, and (4) part of the association between dietary composition and behavioral disinhibition is mediated by cortical and subcortical brain volumes, particularly by frontal-parietal and subcortical areas. 

## 2. Methods

### 2.1. Sample

UK (United Kingdom) Biobank is an open-access, population-based cohort study consisting of in-depth biomedical and health information, as described in [44] or https://www.ukbiobank.ac.uk/ (29 July 2019). Of the full UKBiobank sample, all subjects were included who had segmented T1 MRI data available, as well as the Mental Health Questionnaire (MHQ), the Food Frequency Questionnaire (FFQ), and self-reported fitness measures. Both MRI measurement and MHQ assessment were obtained in a subsample of the full UKBiobank cohort. Importantly, the subsample of UKbiobank with MRI measurement available was designed to be a random selection of the full cohort and did not differ on major demographic variables. A detailed flowchart and description of the subsample inclusion and exclusion criteria can be found [45]. The sample used in the current analyses included a total of 15,258 subjects for which both MRI and MHQ assessments were available (mean age = 55, 7037 males). All analyses were based on a copy of the UKB data downloaded on 29 July 2019.

### 2.2. Behavioral Disinhibition

Behavioral disinhibition is not a standard measure available in the UKBiobank. Hence, we used a single aggregated measure for disinhibition that has been calculated on a much larger sample of the UKBiobank by [22]. In that study, a principal component analysis (PCA) was performed on all available disinhibition-related items from different questionnaires (e.g., items covering addictions including smoking, risk behaviors such as heavy drinking, self-reported and hospital diagnoses of mental health disorders, self-harm behaviors, and personality questionnaire items, among others. A full list of items is available in Appendix A). To obtain a balanced representation of different manifestations of disinhibition, these items were grouped into nine types of disinhibited behaviors. Schweren et al. [22] performed the PCA based on tetrachoric correlations between these behavioral groups using the psych package in R ( Appendix A). The single-component model, preferred a priori, presented with no interpretational shortcomings: all behaviors loaded positively on the principal component with factor loadings ranging from 0.335 to 0.708 (Appendix A). For each subject, a factor score was extracted, with higher scores indicating a higher tendency for disinhibition. We did not re-run this analysis on our subsample but used the factor scores from the original PCA.

### 2.3. Dietary Components

Dietary composition parameters per subject were obtained from the [22] paper, where dietary composition was calculated based on the 29 items of the Food Frequency Questionnaire (FFQ), a self-report questionnaire assessing the participants’ past-year average food consumption. Individual items of the FFQ are highly correlated, reflecting underlying dietary patterns. To derive these patterns, Schweren et al. [22] performed a PCA with Promax rotation implemented in the psych package in R. The PCA started with one principal component, and new components were added one-by-one. Components were retained when they contributed unique information, were interpretable and plausible, and remained stable upon including additional components to the model. The optimal model contained four dietary components. These were used as input for a PCA, rendering four dietary components: Diet PC1 was associated with healthy foods including vegetables, fruits, whole grain bread, and oily fish, which we will therefore label as the ‘healthy diet’. Diet PC2 reflected specific restrictions in bread, milk, and wheat intake, which will therefore be labeled ‘restricted diet’. Diet PC3 was associated with meat and fish consumption, labeled the ‘meat/fish diet’. Diet PC4 was associated with a specific intake of high-fat dairy products, labeled the ‘high-fat dairy diet’. We did not re-calculate these data in our smaller subsample but used the factor loadings from [22] as input in our analyses (see Appendix A for additional details on FFQ items and PCA factor loadings).

### 2.4. Structural Brain Measures

For all subjects with available MRI scans in UKBiobank, T1 images have been automatically segmented by UK Biobank based on the Harvard–Oxford atlas using FLS-FIRST. Cortical volume measures of the resulting 47 cortical regions of interest (ROIs) were included, as well as those of the 6 subcortical segmentations (amygdala, caudate, hippocampus, pallidum, putamen, and thalamus). For all ROIs, left and right hemisphere volumes were averaged to obtain one input measure per region. Outliers were removed at ±3×SD. 

### 2.5. Independent Component Analysis of Structural Brain Measures

To reduce the number of brain measures while maintaining meaningful mediators, we carried out data-reduction in the form of an Independent Component Analysis (ICA), using the z-transformed of the 53 standard subcortical and cortical volume measures as input. 

The number of components for this ICA was started at 2, and components were added based on their interpretability, uniqueness, and stability, with the underlying assumption that all resulting ICs should be meaningful as input in the upcoming mediation analysis. This led to the final selection of an ICA with 4 components since this selection included maximally dissociable individual PCs without including any ICs that could not be functionally interpreted (see Table 1 for the resulting factor loadings for the 4 factor ICA solution. See Figure 1A–D for the visualized factor loadings per IC.). 

Independent component 1 (IC1) could be characterized by high loadings in frontal and cingulate areas, as well as around the temporal/parietal cortex. IC2 showed high loadings in frontal and occipital cortices. IC3 was characterized by positive loadings mainly in the subcortical areas, and IC4 showed high negative loadings on frontal/temporal areas, particularly around the superior temporal cortex. 

### 2.6. Covariates

Age, gender, social-economic status (SES) (educational attainment (years of education), total household income, Indices of Multiple Deprivation (IMD), employment (employed/unemployed)) and ethnicity (white/non-white) were used as covariates. To account for a generally healthier lifestyle outside of dietary composition, we also included BMI and moderate-to-vigorous physical activity (MVPA) as covariates, in line with the previous publication by [22]. MVPA was based on the Recent Physical Activity Questionnaire (RPAQ) [46] which included a self-reported number of days per week that subjects performed physical activity, as well as the number of minutes that they performed this activity on these days. Taken together, this gives us the total minutes of MVPA for the last week for each subject. To account for outliers, the highest 2.5% of MVPA values were removed from the analysis. 

### 2.7. Mediation Analyses

Using the LAVAAN package in R, four separate multiple mediation models were constructed (see Figure 2). Each model used one dietary component as the predictor variable, the behavioral disinhibition factor as the dependent variable, and all four brain volume ICs as mediator variables, thereby testing the full mediation of all the brain measures for each dietary component within the same model. We report 4 pathways within each model, namely: pathway A from dietary composition to the brain, path B from the brain to behavioral disinhibition, C between dietary composition and behavioral disinhibition, and D as the indirect pathway, which is part of the total effect on behavioral disinhibition mediated through the brain. For each pathway, we report the standard Beta coefficient to indicate the direction and size of the effect, as well as FDR (false-discovery rate) corrected *p*-values. Within the mediation models, all the above-mentioned covariates were included to correct for these variables on all pathways. 

## 3. Results

### 3.1. Demographic and Lifestyle Factors

The association between demographic factors and dininhibition are displayed in Table 2. Disinhibition was higher in men compared to women (B = 0.145, *p* < 0.001). Disinhibition was significantly associated with younger age (B = −0.204, *p* < 0.001), unemployment (B = 0.23, *p* = 0.014), white ethnicity (B = 0.12, *p* = 0.035), and neighborhood deprivation (B = 0.09, *p* < 0.01). MVPA was associated with higher disinhibition (B=0.029, *p* = 0.003), as was BMI (B = 0.031, *p* = 0.002). Disinhibition was not associated with adjusted income and years of education. 

### 3.2. Multiple Mediation Models

The first multiple mediation model used healthy diet as the predictor and IC1–4 as the mediator variables (see Table 3). This model showed a significant association between healthy diet and all four brain ICs (range B estimates A1–A4 = −1.05–0.064; *p*-values < 0.001), as well as a significant association between brain IC1 and disinhibition (B estimate B1 = 0.038; *p* < 0.001). Healthy diet was significantly negatively associated with disinhibition (B estimate C = −0.035; *p* < 0.001). Part of the total effect of this model was mediated through the brain IC1 (B estimate D1 = −0.004, *p* < 0.001, 10%). None of the other brain ICs mediated the association between healthy diet and disinhibition (Table 3). 

The second mediation model used restricted diet as the predictor and IC1-4 as the mediator variables (see Table 4). Here we observed a significant association between restricted diet and brain IC3 and IC4 (B estimates A3 = 0.029; *p* < 0.001. A4 = 0.029, *p* < 0.01). The association between restricted diet and disinhibition was positive and significant (B estimate C = 0.018; *p* < 0.01). No part of the total effects was mediated through the brain ICs.

The third mediation model used meat/fish diet as the predictor and IC1–4 as the mediator variables (see Table 5). In this model we observe significant associations between meat/fish diet and brain ICs 1, 2, and 4 (B estimate A1 = 0.052; *p* < 0.001; A2 = −0.058; *p* < 0.001; A4 = −0.148; *p* < 0.001). Meat/fish diet is significantly positively associated with disinhibition (B estimate C = 0.018, *p* < 0.021). Part of the effect in this model is mediated through brain IC1 (B estimate D1 = 0.002; *p* < 0.001). 

The fourth mediation model used high-fat dairy diet as the predictor and IC1–4 as the mediator variables (see Table 6). High-fat dairy diet was associated with IC1 (B estimate A1 = 0.048; *p* < 0.001), IC3 (B estimate A3 = −0.042; *p* < 0.001), and IC4 (B estimate A4 = −0.035; *p* < 0.001). High-fat dairy diet did not significantly associate with disinhibition in this model, though there was a significant indirect effect through IC1 (B estimate D1 = 0.002, *p* < 0.002). 

## 4. Discussion

In this study, we investigated the associations between dietary composition, cortical and subcortical brain volume, and behavioral disinhibition. Our models showed that: (1) there are significant associations between all dietary components and brain volume components; (2) brain volumes (for the first three components) are associated with behavioral disinhibition; (3) the mediation models show that part of the variance in behavioral disinhibition explained by dietary components (for healthy diet, restricted diet, and high-fat dairy diet) is mediated through the frontal-temporal/parietal brain volume component. These results partially confirm our hypotheses on the relation between dietary composition, frontal-parietal brain volume, and behavioral disinhibition. 

The first step in our analyses was an ICA of the cortical and subcortical brain volumes in order to reduce the input data for our mediation models. This analysis resulted in four uniquely interpretable and stable brain components, namely a frontal-parietal-temporal component (IC1), a frontal-occipital component (IC2), a subcortical component (IC3), and a superior temporal component (IC4). The frontal-parietal-temporal and subcortical ICs were the primary components that we hypothesized to associate with both diet and disinhibition. In fact, IC1 was the only brain component associated with disinhibited behavior, the positive association indicating that higher volumes in the frontal-temporal/parietal areas were associated with higher disinhibition scores, which was surprising as we had expected overall lower brain volumes to relate to disinhibition based on previous literature [37,38]. IC1 (frontal-parietal-temporal volume) was also the brain component most strongly associated with all dietary components except the restricted diet. Interestingly, larger brain volumes of IC1 were negatively associated with a healthy diet but positively with meat/fish and high-fat dairy intake. IC2 (frontal-occipital volumes) was negatively associated with the meat/fish diet, and IC3 (subcortical volumes) was positively associated with the healthy diet and restricted diet but negatively with the high-fat dairy diet. IC4 (superior temporal volume) was associated with all diets, showing a positive association with a healthy and restricted diet but a negative association with meat/fish and high-fat dairy intake. 

Our mediation models show that for healthy diet, restricted diet, and high-fat dairy diet, part of the association between diet and disinhibition is mediated through the frontal-temporal/parietal brain volume component. The healthy diet shows a negative association with disinhibition, suggesting an overall protective effect of a healthy diet, of which part may be due to the influence of a healthy diet on frontal-temporal/parietal brain volumes. For the restricted diet, no part of the positive association between dietary restrictions and disinhibition was mediated through the brain. For the meat/fish diet, there was also a positive association between diet and disinhibition, indicating more meat/fish consumption was associated with more disinhibitory behavior, part of which was mediated through the association between more meat/fish consumption and higher frontal-temporal/parietal volumes. For the high-fat dairy diet, we could not replicate the direct effect between diet and disinhibition observed by [22], meaning that although an indirect effect was statistically significant, this was not meaningfully interpretable. 

Overall, the results above indicate a primary role for the frontal-parietal/temporal network in linking diet and disinhibition, which is in line with both inhibition literature and our hypotheses that the frontal cognitive control areas would be involved. This is the first study to show such mediation within a large general population sample. No subcortical effects were discovered, suggesting that higher cortical areas might be more sensitive than subcortical ones to dietary effects. Our mediation models further show that only a small portion of the association between diet and disinhibition is mediated through brain volumes. We expect that more detailed volumetric measures (i.e., voxel-based analyses) or functional brain imaging (fMRI or resting-state fMRI) may explain further parts of this variance. However, we must also remain open to other mediating factors beyond brain metrics that may explain parts of the links between diet and disinhibition. Biological factors like inflammation [47] or (epi)genetics [48,49] have also been linked to both diet and inhibition and may explain further parts of the association. 

The disinhibition score that we used in this study was based on a mixed set of interview questions, online questionnaires, and general health records. The resulting disinhibition scale represents impulsive, compulsive, and emotionally unstable features [22]. It can be interpreted as an overarching impulsivity construct linked to impulsive behavior within different psychiatric disorders. This interpretation fits well within the categorical interpretation of mental health constructs within the RDoC approach [50]. Previously, we showed that the disinhibition score was associated in the expected directions with gender, age, socioeconomic status, and MPVA [22]; see also above for the current subsample), and both in the previous work and the current study, disinhibition was found to be correlated with dietary patterns. It must be noted that within this cross-sectional study, we can only observe associations between covariates, diet, the brain, and disinhibition and are unable to make claims about causality. Our assumption, though, reflected in the way we have set up our mediation models, is that diet influences the brain, which in turn contributes to disinhibitory behavior. Specifically, several known causal pathways exist through which diet can influence brain functioning. For instance, intake of healthy foods is known to decrease inflammation markers and thereby decrease neuroinflammation [51,52]. Similarly, HPA-axis activation is shown to be downregulated through decreased pro-inflammatory cytokines and glucocorticoid levels in response to dietary alterations. Upregulated histone acetylation and BDNF expressions can also lead to epigenetic changes in response to dietary interventions. In these ways, we postulate that unhealthy dietary composition over the lifespan could lead to altered brain structure and functioning, which may subsequently lead to issues like increased disinhibitory behavior. Further studies in the UKBiobank and other samples should take these pathways into account to further complete the causal relations between diet, disinhibition, and the brain. 

However, as shown in the Appendix A, these mediation models can equally validly be turned around. Indeed, it is plausible and has been shown in scientific studies that impulsivity may influence food choices [53,54,55]. More insight into the causal directions underlying the association between diet, brain, and disinhibition, therefore, requires a large-scale longitudinal study, tracking these variables over development. Intervention studies where diet is experimentally manipulated would also be able to shed light on these causal pathways. 

### Limitations

Apart from the cross-sectional nature of the UKBiobank cohort described above, there are several limitations inherent in the current study. First, the UKBiobank project was not aimed at indexing disinhibition, meaning our inhibition measure is not easily comparable to other studies using impulsive populations, like subjects with ADHD. The flip-side of the argument is that we used a transdiagnostic construct, it is unlikely that individual differences in diet and brain volume are associated with specific psychiatric conditions. Another limitation lies in the unknown presence of confounding variables. We carefully selected the variables we assumed a priori to potentially confound associations between diet, disinhibition, and brain volumes, but without experimental intervention, we cannot rule out unmeasured confounding that may influence the observed associations. Particularly, factors like medication use and the presence of specific medical disorders may influence the relation between diet, disinhibition, and the brain. We suggest that further analyses of the UKBiobank dataset should be aimed at discerning additional influential factors in this framework. The nature of the UKBiobank cohort reflects only an older part (40+ years) of the general population of adults. In this cohort, we find overall small effects of the general dietary pattern on brain and behavior, but it might be the case that specific subgroups or specific age ranges are more sensitive to dietary effects. The publication by [22] shows differential associations between dietary composition and disinhibition for men and women. Given that gender influences both brain structure and disinhibitory behavior, one of the main directions of future research should be regarding the influence of gender on the link between diet, brain, and disinhibition.

## 5. Conclusions

To conclude, this study shows that frontal-parietal brain volumes partly mediate the association between dietary composition and behavioral disinhibition. Given the strong association between behavioral disinhibition and functional brain alterations, we recommend that further research investigate a wider range of structural and functional brain measures. We further postulate that large-scale dietary intervention, as well as longitudinal observational studies, may be necessary to definitively identify the causal relations between diet, brain, and behavioral disinhibition. Additionally, part of the association between diet and disinhibition that is not mediated directly by neural factors may be explained by other biological factors like inflammation markers or (epi)genetics, which would also be prospective targets for future research.

## Figures and Tables

**Figure 1 nutrients-13-03542-f001:**
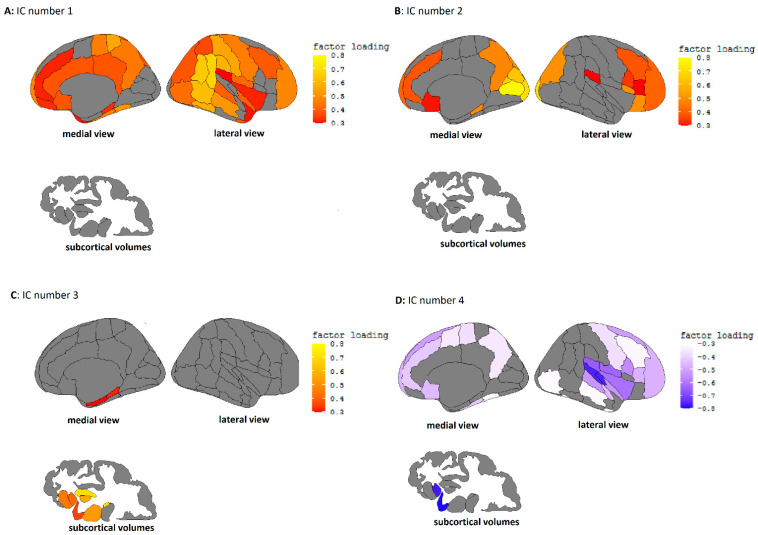
(**A**). Independent component 1, characterized by high factor loadings in the temporal/parietal and frontal cortex. (**B**). Independent component 2, characterized by high loadings in the occipital and frontal cortex. (**C**). Independent component 3, characterized by high loading in the temporal cortex and subcortical volumes. (**D**). Independent component 4, characterized by high loadings in the temporal cortex.

**Figure 2 nutrients-13-03542-f002:**
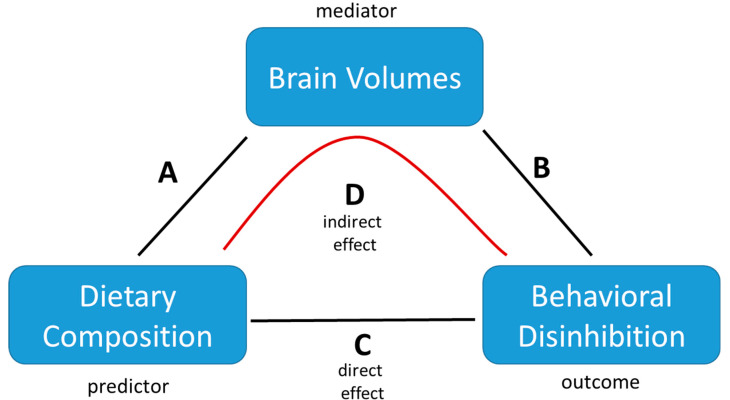
Schematic illustration of the multiple mediation model employed to analyze the associations between dietary composition, brain volumes, and behavioral disinhibition. The model tests for the significance of any of the direct effects (**A**–**C**) as well as whether the indirect effect of (**D**) explains (part of) the total effect. Covariates are not depicted but were included in all analyses (**A**–**D**) as described in the text.

**Table 1 nutrients-13-03542-t001:** Factor loadings of all cortical and subcortical structural brain volume segmentations resulting from Independent Component Analysis. Loadings higher than 0.3 are indicated in **bold** and are plotted in Figure 1A–D.

Frontal Cortex	IC1	IC2	IC3	IC4
frontal pole	**0.48**	**0.41**	0.11	**−0.47**
superior frontal gyrus	0.21	0.22	0.09	**−0.51**
medial frontal gyrus	**0.39**	**0.38**	0.06	**−0.32**
inferior frontal gyrus, ant	−0.05	**0.34**	−0.04	**−0.55**
inferior frontal gyrus, post	0.15	**0.30**	−0.09	**−0.47**
frontal medial cortex	**0.33**	0.14	−0.05	−0.24
juxtrapositional lobule cortex	0.18	0.13	0.04	**−0.35**
frontal orbital cortex	0.05	**0.34**	−0.04	**−0.62**
precentral gyrus	0.46	0.28	0.04	**−0.37**
oribitofrontal cortex	0.24	**0.51**	0.14	−0.25
central oppercular cortex	**0.42**	0.27	−0.01	**−0.63**
posterior opercular cortex	**0.30**	**0.30**	0.06	**−0.65**
Insula				
insular cortex	**0.32**	0.26	0.27	**−0.61**
cingulate cortex				
subcallossal cortex	**0.38**	0.31	0.20	**−0.45**
paracingulate gyrus	**0.32**	0.38	0.01	**−0.42**
cingulate cortex	**0.38**	0.27	0.21	−0.14
cingulate cortex	**0.51**	0.29	0.31	**−0.31**
parietal cortex				
postcentral gyrus	**0.56**	0.27	−0.05	−0.26
superior parietal lobule	**0.36**	0.19	0.03	−0.21
supramarginal gyrus, anteriordivision	**0.60**	0.17	0.00	−0.21
supramarginal gyrus, posteriordivision	**0.68**	0.15	0.05	−0.15
angular gyrus	**0.65**	0.11	0.07	−0.10
occipital cortex				
lateral occipital cortex, inf	**0.43**	0.29	0.14	**−0.31**
lateral occipital cortex, sup	**0.38**	**0.51**	0.10	−0.24
intracalcine cortex	0.00	**0.78**	0.18	−0.09
precuneus	**0.45**	**0.48**	0.14	**−0.35**
cuneus	0.25	**0.64**	0.00	−0.23
lingual cortex	**0.33**	**0.49**	**0.32**	−0.22
occipital fusiform gyrus	0.24	**0.51**	0.14	−0.25
supracalcine cortex	0.29	**0.67**	0.12	−0.12
occipital pole	0.14	**0.71**	0.09	−0.24
temporal cortex				
temporal pole	**0.31**	0.18	0.15	−0.22
superior temporal gyrus, ant	0.26	0.09	0.10	**−0.45**
superior temporal gyrus, post	**0.40**	0.17	0.12	**−0.53**
medial temporal gyrus, ant	0.20	0.11	0.16	−0.26
medial temporal gyrus, post	**0.52**	0.18	0.14	**−0.31**
medial temporal gyrus, temp	**0.62**	0.24	0.12	−0.10
inferior temporal gyrus, ant	0.18	0.03	0.15	−0.20
inferior temporal gyrus, post	**0.39**	0.11	0.08	**−0.30**
inferior temporal gyrus, temp	**0.58**	0.18	0.09	−0.19
parahippocampal gyrus	0.28	0.13	**0.30**	−0.14
parahippocampal gyrus, post	0.24	0.17	0.27	−0.23
temporal fusiform cortex, anterior	**0.31**	0.03	0.19	−0.21
temporal fusiform cortex, posterior	**0.47**	0.08	0.12	**−0.42**
temporal occipital cortex	**0.52**	0.16	0.23	**−0.31**
planum polare	**0.40**	0.20	0.07	**−0.53**
heschl’s gyus	0.21	0.29	0.00	**−0.76**
planum temporale	0.22	0.27	0.09	**−0.76**
subcortical areas				
thalamus	0.19	0.20	**0.53**	−0.17
caudate	0.15	0.16	**0.70**	−0.10
pallidum	−0.02	0.01	**0.57**	0.05
hippocampus	0.25	0.22	**0.45**	−0.27
amygdala	0.24	0.24	**0.42**	**−0.33**
putamen	0.02	0.12	**0.71**	−0.08
nucleus accumbens	−0.17	0.09	**0.35**	−0.31

**Table 2 nutrients-13-03542-t002:** Sample characteristics and associations with Behavioral Disinhibition. Unemployment refers to current employment status (0 = no employment, 1 = currently employed). Ethnicity was coded as 0 = white, 1 = non-white. IMD = Indices of Multiple Deprivation. MVPA = Moderate/Vigorous Physical Activity.

Sample	Mean	Association with Disinhibition (B)	*p*-Value
Sex	7037 m 8221f	0.145	<0.001
Age	40–69 yo mean = 55 yo	−0.204	<0.001
Unemployment		0.23	0.014
Ethnicity		0.12	0.035
IMD		0.09	<0.01
MVPA		0.029	0.003

**Table 3 nutrients-13-03542-t003:** Multiple mediation model for the healthy diet component. Bold values indicate significant associations.

Healthy Diet						
Effect		B	Std. Error	z-Value	*p* (Adjusted)	% of Total Effect
A1	Healthy Diet - IC1	**−0.105**	**0.008**	**−12.767**	**<0.001**	
A2	Healthy Diet - IC2	**0.064**	**0.008**	**7.828**	**<0.001**	
A3	Healthy Diet - IC3	**0.019**	**0.008**	**2.375**	**<0.018**	
A4	Healthy Diet - IC4	**0.047**	**0.008**	**5.718**	**<0.001**	
B1	IC1 - Disinhibition	**0.038**	**0.008**	**4.487**	**<0.001**	
B2	IC2 - Disinhibition	−0.012	0.008	−1.532	0.126	
B3	IC3 - Disinhibition	−0.012	0.008	−1.563	0.118	
B4	IC4 - Disinhibition	−0.009	0.008	−1.129	0.259	
C	Healthy Diet - Disinhibition	**−0.035**	**0.008**	**−4.3**	**<0.001**	87.5
D1	Healthy Diet - IC1 - Disinhibition	**−0.004**	**0.001**	**−4.23**	**<0.001**	10
D2	Healthy Diet - IC2 - Disinhibition	−0.001	0.0001	−1.493	0.135	2.5
D3	Healthy Diet - IC3- Disinhibition	−0.0001	0.0001	−1.183	0.237	0.25
D4	Healthy Diet - IC4 - Disinhibition	−0.0001	0.0001	−1.094	0.274	0.25
TOTAL	C + D (direct + indirect effects)	**−0.040**	**0.008**	**−4.915**	**<0.001**	

**Table 4 nutrients-13-03542-t004:** Multiple mediation model for the restricted diet component. Bold values indicate significant associations.

Restricted Diet						
Effect		B	Std. Error	z-Value	*p* (Adjusted)	% of Total Effect
A1	Restricted Diet – IC1	0	0.009	0.033	0.974	
A2	Restricted Diet – IC2	−0.016	0.009	−1.723	0.085	
A3	Restricted Diet – IC3	**0.029**	**0.008**	**3.617**	**<0.001**	
A4	Restricted Diet – IC4	**0.029**	**0.01**	**3.002**	**<0.003**	
B1	IC1 – Disinhibition	**0.039**	**0.008**	**4.669**	**<0.001**	
B2	IC2 – Disinhibition	−0.012	0.009	−1.347	0.178	
B3	IC3 – Disinhibition	−0.012	0.007	−1.768	0.077	
B4	IC4 – Disinhibition	−0.01	0.007	−1.376	0.169	
C	Restricted Diet - Disinhibition	**0.018**	**0.007**	**2.56**	**<0.01**	105.8
D1	Restricted Diet - IC1 - Disinhibition	0	0	0.03	0.976	0
D2	Restricted Diet - IC2 - Disinhibition	0	0	0.96	0.337	0
D3	Restricted Diet - IC3- Disinhibition	0	0	−1.355	0.175	0
D4	Restricted Diet - IC4 - Disinhibition	0	0	−1.139	0.255	0
TOTAL	C + D (direct + indirect effects)	0.017	0.007	2.447	0.014	

**Table 5 nutrients-13-03542-t005:** Multiple mediation model for the meat/fish diet component. Bold values indicate significant associations.

Meat/Fish Diet						
Effect		B	Std. Error	z-Value	*p* (Adjusted)	% of Total Effect
A1	Meat/Fish Diet – IC1	**0.052**	**0.008**	**6.725**	**<0.001**	
A2	Meat/Fish Diet – IC2	**−0.058**	**0.008**	**−6.888**	**<0.001**	
A3	Meat/Fish Diet – IC3	−0.016	0.008	−1.851	0.064	
A4	Meat/Fish Diet – IC4	**−0.148**	**0.008**	**−17.864**	**<0.001**	
B1	IC1 – Disinhibition	**0.039**	**0.007**	**5.293**	**<0.001**	
B2	IC2 – Disinhibition	−0.013	0.008	−1.509	0.131	
B3	IC3 – Disinhibition	−0.012	0.007	−1.712	0.087	
B4	IC4 – Disinhibition	−0.009	0.007	−1.357	0.175	
C	Meat/Fish Diet - Disinhibition	**0.018**	**0.008**	**2.313**	**<0.021**	0.78
D1	Meat/Fish Diet - IC1 - Disinhibition	**0.002**	**0.001**	**3.826**	**<0.001**	0.09
D2	Meat/Fish Diet - IC2 - Disinhibition	0.001	0	1.476	0.14	0.04
D3	Meat/Fish Diet - IC3- Disinhibition	0	0	1.114	0.265	0.00
D4	Meat/Fish Diet - IC4 - Disinhibition	0.001	0.001	1.345	0.179	0.04
TOTAL	C + D (direct + indirect effects)	0.023	0.008	2.782	0.005	

**Table 6 nutrients-13-03542-t006:** Multiple mediation model for the high-fat dairy diet component. Bold values indicate significant associations.

High-Fat Dairy						
Effect		B	Std. Error	z-Value	*p* (Adjusted)	% of Total Effect
A1	High-fat dairy Diet – IC1	**0.048**	**0.008**	**5.592**	**<0.001**	
A2	High-fat dairy Diet – IC2	−0.023	0.008	−2.732	0.006	
A3	High-fat dairy Diet – IC3	**−0.042**	**0.008**	**−5.493**	**<0.001**	
A4	High-fat dairy Diet – IC4	**−0.035**	**0.008**	**−4.375**	**<0.001**	
B1	IC1 – Disinhibition	**0.039**	**0.009**	**4.212**	**<0.001**	
B2	IC2 – Disinhibition	−0.013	0.008	−1.623	0.105	
B3	IC3 – Disinhibition	−0.012	0.008	−1.446	0.148	
B4	IC4 - Disinhibition	−0.01	0.008	−1.352	0.176	
C	High-fat Diet - Disinhibition	0.012	0.007	1.788	0.074	0.80
D1	High-fat Diet - IC1 - Disinhibition	**0.002**	**0.001**	**3.057**	**<0.002**	0.13
D2	High-fat Diet - IC2 - Disinhibition	0	0	1.37	0.171	0.00
D3	High-fat Diet - IC3- Disinhibition	0.001	0	1.34	0.18	0.07
D4	High-fat Diet - IC4 - Disinhibition	0	0	1.316	0.188	0.00
TOTAL	C + D (direct + indirect effects)	0.015	0.007	2.17	0.03	

## Data Availability

Bona fide researchers can apply to access the UK Biobank research resource to conduct health-related research that is in the public interest. Access to the data was granted by UK Biobank following registration with their system and approval of our research project (project number 23668). All derived variables (behavioral disinhibition, four dietary components, and one dietary grouping variable) will be returned and shared through UK Biobank. For details, see https://www.ukbiobank.ac.uk/ (accessed on 29 July 2019).

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
