# Peer review of "Cortical and Subcortical Brain Volumes Partially Mediate the Association between Dietary Composition and Behavioral Disinhibition: A UK Biobank Study"

_nutrients, 2021, doi:10.3390/nu13103542_

Round 1
Reviewer 1 Report
The manuscript is well written, and the study is very interesting.
I have few comments and suggestions for the authors
- Please provide more information about PCA. How did you determine the number of factors, what was the cut-off of significant loading? It appears to me that the components are very diluted. I would expect for a healthy diet to see more than fruits, vegetables, and healthy oils.
- Please define “healthy oils” in the PC1.
- The figures labeled “subcortical” (in figure 2) seem to me similar between IC 1 and IC2. It also very hard to read the legend.
- Please provide a rationale for choosing the 2.5 % of MVPA to remove outliers.
- Please provide the results described in section 3.1 in a table with Coefficients and P values, which will make the paragraph less cumbersome to read.
- It would be interesting if the discussion includes connection between the dietary findings, brain structures and disinhibition. It will make the study more meaningful.
- Although I feel this a very interesting study, the fact that sexes and different ages ( 40+) were all combined concerns as some evidence suggests different cortical volumes between men and women as well as depending on age and lifestyle factors ( smokers, drinkers vs active and healthy). The authors did address some of these limitations , but they may need to point out the sex and the different lifestyle factors as confounding factors as well.
- It would be nice to breakdown the sections into sub-sections which will make the manuscript easier to read.
Author Response
Reviewer 1:
1. Please provide more information about PCA. How did you determine the number of factors, what was the cut-off of significant loading? It appears to me that the components are very diluted. I would expect for a healthy diet to see more than fruits, vegetables, and healthy oils.
Response: we agree with the reviewer that the description of the PCA analyses are not sufficiently detailed, and relies too much on the Schweren et al. paper which will be published in the same issue of Nutrients. Therefore, we have added several supplementary tables to describe the generation of the Diet PC’s (Tables SI1 and SI2). The section in the methods has been adapted as follows:
“2.3. Dietary components
Dietary composition parameters per subject were obtained from the Schweren et al. (2021) paper, where dietary composition was calculated based on the 29 items of the Food Frequency Questionnaire (FFQ), a self report questionnaire assessing the participants’ past-year average food consumption. Individual items of the FFQ are highly correlated, reflecting underlying dietary patterns. To derive these patterns, Schweren et al. performed a PCA with promax rotation implemented in the psych package in R. The PCA started with one principal component, and new components were added one-by-one. Components were retained when they contributed unique information, were interpretable and plausible, and remained stable upon including additional components to the model. The optimal model contained four dietary components.These were used as input for a PCA, rendering four dietary components: Diet PC1 was associated with healthy foods including vegetables, fruits and healthy oils, which we will therefore label as the ‘healthy diet’. Diet PC2 reflected specific restrictions in bread, milk and wheat intake, which will therefore be labelled ‘restricted diet’. Diet PC3 was associated with meat and fish consumption, labelled the ‘meat/fish diet’. Diet PC4 was associated with specific intake of high-fat dairy products, labelled the ‘high-fat dairy diet’. We did not to re-calculate these data in our smaller subsample, but used the factor loadings from Schweren et al. (2021) as input in our analyses (see the Supplementary Table SI1 and SI2 for additional details on FFQ items, and PCA factor loadings).”
2. Please define “healthy oils” in the PC1.
Response: we adapted the description of PC1 a bit to reflect more directly the items loading on this factor (see Table SI1):
“Diet PC1 was associated with healthy foods including vegetables, fruits and wholegrain bread and oily fish, which we will therefore label as the ‘healthy diet’.”
3. The figures labeled “subcortical” (in figure 2) seem to me similar between IC 1 and IC2. It also very hard to read the legend.
Response: they are indeed identical, as neither IC1 or IC2 load significantly on any of the subcortical areas. We have adapted figure 2 to be more easily readable.
4. Please provide a rationale for choosing the 2.5 % of MVPA to remove outliers.
Response: This cutoff for ourlier detection is based on the distribution of the data. Since MVPA is highly non-normally distributed, a cutoff based on standard deviations makes less sense. In particular, we observe only outliers in the top of the distribution, which are likely to do reporting error. Hence, as an alternative, we have chosen to manually select the top 2.5% instead of relying on 2xSD. This is a similar method used for different data types in the Schweren et al., 2021 paper on which the current manuscript is based. In order to keep methodological similarities between our two manuscripts as high as possible, we opted to keep this same cutoff for MVPA in the current analyses.
5. Please provide the results described in section 3.1 in a table with Coefficients and P values, which will make the paragraph less cumbersome to read.
Response: In line with this suggestions, we have added a new table describing the main demographic factors and some of the important covariate effects in section 3.1
6. It would be interesting if the discussion includes connection between the dietary findings, brain structures and disinhibition. It will make the study more meaningful.
Response: we agree with the reviewer that we could be more explicit in the causal relations that we believe could underlie the associations found in the current study. we have added the following section to the discussion:
“Specifically, several known causal pathways exist through which diet can influence brain functioning. For instance, intake of healthy foods is known to decrease inflammation markers, and thereby decrease neuroinflammation (Wang et al., 2016, Johnson et al., 2020). Similarly, HPA-axis activation is shown to be downregulated though decreased proinflammatory cytokines and glucocorticoid levels in response to dietary alterations. Upregulated histone acetylation and BDNF expressions can also lead to epigenetic changes in response to dietary interventions (Johnson et al., 2020). ). In these ways, we postulate that unhealthy dietary composition over the lifespan could lead to altered brain structure and functioning, which may subsequently lead to issues like increased disinhibitory behavior. Further studies in the UKBiobank and other samples should take these pathways into account to further complete the causal relations between diet, disinhibition and the brain.”
7. Although I feel this a very interesting study, the fact that sexes and different ages ( 40+) were all combined concerns as some evidence suggests different cortical volumes between men and women as well as depending on age and lifestyle factors ( smokers, drinkers vs active and healthy). The authors did address some of these limitations , but they may need to point out the sex and the different lifestyle factors as confounding factors as well.
Response: we agree with the reviewer that several other factors are of major importance for the current study. The current results are aimed to find associations in as general a sample as possible, while follow up studies are underway to tease out the effect of specific other influential factors. Sex, like other lifestyle and SES factors are therefore only taken as covariates in our models, but it's influence is not studies here in any detail. However, as sex has a profound influence on disinhibition and brain development, and we adapted the limitations section of the discussion to acknowledge this as follows:
“Apart from the cross-sectional nature of the UK biobank cohort described above, there are several limitations inherent in the current study. First, the UK biobank project was not aimed at indexing disinhibition, meaning our inhibition measure is not easily comparable to other studies using impulsive populations, like subjects with ADHD. The flip-side of the argument is that we used transdiagnostic construct – it in unlikely that individual differences in diet and brain volume are associated with specific psychiatric conditions. Another limitation lies in the unknown presence of confounding variables. We carefully selected the variables which we assumed a-priori to potentially confound associations between diet, disinhibition and brain volumes, but without experimental intervention we cannot rule out unmeasured confounding that may influence the observed associations. Particularly, factors like medication use and the presence of specific medical disorders may influence the relation between diet, disinhibition and the brain. We suggest further analyses of the UK biobank dataset should be aimed at discerning additional influential factors in this framework. The nature of the UKBiobank cohort reflects only an older part (40years+) of the general population of adults. In this cohort, we find overall small effects of the general dietary pattern on brain and behavior, but it might be the case that specific subgroups or specific age ranges are more sensitive to dietary effects. The publication by Schweren et al., (2021) shows differential associations between dietary composition and disinhibition for men and women. Given that sex influence both brain structure and disinhibitory behavior, one of the main direction s of future research should be regarding he influence of sex on the link between diet, brain and disinhibition.”
8. It would be nice to breakdown the sections into sub-sections which will make the manuscript easier to read.
Response: We have the Nutrients house style for layout, including sub-sections in the methods and results section. In response to the reviewer suggestion, have now also added sub-sections for the limitations and conclusions section in the discussion.

Reviewer 2 Report
Thank you for submitting the manuscript entitled "Cortical and subcortical brain volumes partially mediate the association between dietary composition and behavioral disinhibition: a UK Biobank study"
The article is very promising, it studies the relationship between diet and disinhibition behavior and how brain volume could mediate this relationship. To this purpose, the authors study a series of multiple mediation models. This article provides new findings in this field and may be of interest to readers. In subsequent sections, I suggest a few considerations to the authors and ask them a series of questions that have emerged to me while reading the article:
- Watch for spaces, for example "Table3" without spaces.
- Keep an eye on the capital letters in the tables.
- Keep an eye on the 0’s, sometimes they are included and sometimes they are not.
- Line numbers are not included.
- Check that the abbreviations used have been explained before or are at the end of the manuscript. g. ROI.
ABSTRACT:
I would include in the abstract why the results obtained are important.
METHODS:
- I suggest to include a flow-chart of subsample selection. I suggest that the criteria for inclusion and exclusion of the participants in the subsample be exposed.
- Why are these dietary patterns analyzed and not others? This point of the methods is a little short for me and I think the section should be better explained. I have not been able to see the supplementary material.
- I advise to provide how the FFQ was collected, if it was self-completed, at what time of the year...Also better describe the foods that make up each of the food patterns studied, just as the components of the brain measures are described. It would be nice to know what foods make up each dietary pattern to get an idea of the nutrients they may be taking in each consumption pattern. For example, does the high fat pattern include all types of fats (saturated fats or polyunsaturated fats) or what type of fat is prevalent in this food pattern.
- Can you provide more information about the biobank? For example, a reference website.
- Specify all the items of the behavioral disinhibition score. A scheme for obtaining the score could be included with all the variables that have been considered.
- Why have other variables that could be affecting the relationship studied not been taken into account, for example, smoking history, the presence of diseases or the taking of medications that can affect behavior.
- Explain how the main component analysis was carried out for both Behavioral disinhibition and dietary variables. Was it performed in R? What statistical package was used? Can you cite with references the statistical analyses used?
RESULTS:
- Why not describe the sample in a comparative table setting out its main characteristics?
- Maybe, Figure 1 be better understood if they used arrows instead of straight lines and included which variable is the predictor, dependent and mediating variables, as well as which are direct and indirect relationships?
DISCUSSION:
I think could be improved by further discussing some of the concepts it deals with.
- Could you hypothesize the mechanism of how diet could affect to behavioral disinhibition? For example, there are nutrients as salt, sugars of fat acids that could affect an impulsive or compulsive features or how impulsivity may influence food choices?
- Why do you think inflammation markers or epigenetics could affect to this association?
Author Response
Reviewer 2:
Thank you for submitting the manuscript entitled "Cortical and subcortical brain volumes partially mediate the association between dietary composition and behavioral disinhibition: a UK Biobank study"
The article is very promising, it studies the relationship between diet and disinhibition behavior and how brain volume could mediate this relationship. To this purpose, the authors study a series of multiple mediation models. This article provides new findings in this field and may be of interest to readers. In subsequent sections, I suggest a few considerations to the authors and ask them a series of questions that have emerged to me while reading the article:
- Watch for spaces, for example "Table3" without spaces.
- Keep an eye on the capital letters in the tables.
- Keep an eye on the 0’s, sometimes they are included and sometimes they are not.
- Line numbers are not included.
- Check that the abbreviations used have been explained before or are at the end of the manuscript. g. ROI.
Response: we thank the reviewer for pointing out these issues, the manuscript has been checked and corrected where necessary.
ABSTRACT:
I would include in the abstract why the results obtained are important.
Response: we agree with the reviewer that this is an important point, but also feel we need to be careful not to overstate the causal nature of our findings. Therefore, we added the following sentence in the conclusion of the abstract:
“….These results are in part confirming our hypotheses, and offer a first insight into the underlying mechanisms linking dietary composition, frontal-parietal brain volume and behavioral disinhibition in the general adult population.”
METHODS:
- I suggest to include a flow-chart of subsample selection. I suggest that the criteria for inclusion and exclusion of the participants in the subsample be exposed.
Response: the subsample selection process was not a part of the current study. We selected all subjects for which imaging data is available. A flowchart on which subjects were scanned, and their characterization, is available in the referenced UK biobank methods publication on this subjects (Littlejohns et al., 2020), but cannot be replicated as such in our manuscript due to copyright issues.
Flowchart taken from Littlejohns et al., 2020 paper, as referenced in the main manuscript.
We have adapted the sample description as follows, to make this more clear:
“Both MRI measurement and MHQ assessment were obtained in a subsample of the full UKBiobank cohort. Importantly, the subsample of UKbiobank with MRI measurement available was designed to be a random selection of the full cohort, and did not differ on major demographic variables. A detailed flowchart and description of the subsample inclusion and exclusion criteria can be found in (Littlejohns et al., 2020). The sample used in the current analyses included the total 15,258 subjects for which both MRI and MHQ assessments were available (mean age=55, 7,037 males).”
- Why are these dietary patterns analyzed and not others? This point of the methods is a little short for me and I think the section should be better explained. I have not been able to see the supplementary material.
Response: we acknowledge the point of the reviewer that the section on dietary composition was overly short, and relied too much on the availability of the Schweren et al. paper from the same issue of Nutrients to be comprehensible. Hence, we have adapted the section in the methods section as follows, and have added several tables to the supplement with detailed information about the emergence of these dietary patterns:
“2.3. Dietary components
Dietary composition parameters per subject were obtained from the Schweren et al. (2021) paper, where dietary composition was calculated based on the 29 items of the Food Frequency Questionnaire (FFQ), a self report questionnaire assessing the participants’ past-year average food consumption. Individual items of the FFQ are highly correlated, reflecting underlying dietary patterns. To derive these patterns, Schweren et al. performed a PCA with promax rotation implemented in the psych package in R. The PCA started with one principal component, and new components were added one-by-one. Components were retained when they contributed unique information, were interpretable and plausible, and remained stable upon including additional components to the model. The optimal model contained four dietary components.These were used as input for a PCA, rendering four dietary components: Diet PC1 was associated with healthy foods including vegetables, fruits and healthy oils, which we will therefore label as the ‘healthy diet’. Diet PC2 reflected specific restrictions in bread, milk and wheat intake, which will therefore be labelled ‘restricted diet’. Diet PC3 was associated with meat and fish consumption, labelled the ‘meat/fish diet’. Diet PC4 was associated with specific intake of high-fat dairy products, labelled the ‘high-fat dairy diet’. We did not to re-calculate these data in our smaller subsample, but used the factor loadings from Schweren et al. (2021) as input in our analyses (see the Supplementary Table SI1 and SI2 for additional details on FFQ items, and PCA factor loadings).”
- I advise to provide how the FFQ was collected, if it was self-completed, at what time of the year...Also better describe the foods that make up each of the food patterns studied, just as the components of the brain measures are described. It would be nice to know what foods make up each dietary pattern to get an idea of the nutrients they may be taking in each consumption pattern. For example, does the high fat pattern include all types of fats (saturated fats or polyunsaturated fats) or what type of fat is prevalent in this food pattern.
Response: we agree that this is an important point, and have added the full FFQ as supplementary table SI1, as well as the loadings of these items on the dietary PC’s as supplementary table SI2.
- Can you provide more information about the biobank? For example, a reference website.
Response: Although some references to descriptive papers on the UK biobank were already in the manuscript, we agree with the reviewer that these could be expanded upon. We have added a link to the descriptive paper and the website in the first sentence of de sample section of the methods as follows:
“2.1. Sample
UK (United Kingdom) Biobank is an open-access, population-based cohort study consisting of in-depth biomedical and health information, as described in (Bycroft et al., 2018 or https://www.ukbiobank.ac.uk/). “
- Specify all the items of the behavioral disinhibition score. A scheme for obtaining the score could be included with all the variables that have been considered.
Response: This is also essential information. We have included these scores in supplementary tables SI3 and SI4, and have added their loading on the behavioral disinhibition component in supplementary table SI5. The methods section on behavioral disinhibition has been updated as follows:
“2.2. Behavioral disinhibition
Behavioral disinhibition is not a standard measure available in the UKBiobank. Hence, we used a single aggregated measure for disinhibition that has been calculated on a much larger sample of the UKBiobank by Schweren et al., 2021. In that study, a principal component analysis (PCA) was performed on all available disinhibition-related items from different questionnaires (e.g. items covering addictions including smoking, risk behaviours such as heavy drinking, self-reported and hospital diagnoses of mental health disorders, self-harm behaviours and personality questionnaire items, among others. A full list of items is available in the Supplementary Table SI 3-4).To obtain a balanced representation of different manifestations of disinhibition, these items were grouped into nine types of disinhibited behaviors. Schweren et al. performed the PCA based on tetrachoric correlations between these behavioural groups using the psych package in R (Supplementary Table SI5). The single-component model, preferred a priori, presented with no interpretational shortcomings: all behaviors loaded positively on the principal component with factor loadings ranging from 0.335 to 0.708 (Supplementary Table SI5). each subject, a factor score was extracted with higher scores indicating a higher tendency for disinhibition. We did not re-run this analysis on our subsample, but used the factor scores from the original PCA.”
- Why have other variables that could be affecting the relationship studied not been taken into account, for example, smoking history, the presence of diseases or the taking of medications that can affect behavior.
Response: we agree with the reviewer that there are many other factors that would be interesting to take a look at in relation to diet and disinhibition. In the current publication we have opted to include factors related to SES, lifestyle and ethnicity which may influence dietary patterns in the general population. On the other hand, factors like specific mediation effects or the presence of medical or psychiatric problems may strongly influence the association between diet and inhibition in specific cases. However, given the vast amount of medication information in the UK biobank cohort, a full study of all these effects goes far beyond the scope of the current paper.
We have added the following statement in the discussion/limitations section to address this issue:
“Particularly, factors like medication use and the presence of specific medical disorders may influence the relation between diet, disinhibition and the brain. We suggest further analyses of the UK biobank dataset should be aimed at discerning additional influential factors in this framework”
- Explain how the main component analysis was carried out for both Behavioral disinhibition and dietary variables. Was it performed in R? What statistical package was used? Can you cite with references the statistical analyses used?
Response: both PCA’s were performed using the principal function from the psych package in R. This has been added to the methods as follows:
“The optimal model contained four dietary components.These were used as input for a PCA implemented in the psych package in R…”
And
“Schweren et al. performed the PCA based on tetrachoric correlations between these behavioural groups using the psych package in R”
RESULTS:
- Why not describe the sample in a comparative table setting out its main characteristics?
Response: In line with this suggestions, we have added a new table describing the main demographic factors and some of the important covariate effects in section 3.1
- Maybe, Figure 1 be better understood if they used arrows instead of straight lines and included which variable is the predictor, dependent and mediating variables, as well as which are direct and indirect relationships?
Response: the reviewer suggests several interesting additions to figure 1, which we have added below. We will leave out the addition of any arrows, particularly to dissuade any suggestions of causal directionality.
DISCUSSION:
I think could be improved by further discussing some of the concepts it deals with.
- Could you hypothesize the mechanism of how diet could affect to behavioral disinhibition? For example, there are nutrients as salt, sugars of fat acids that could affect an impulsive or compulsive features or how impulsivity may influence food choices?
- Why do you think inflammation markers or epigenetics could affect to this association?
Response: We would like to answer both points in one answer, as we have added a further section in the discussion, filling in the potential causal pathways from diet to disinhibition further. The following section as been added:
“Specifically, several known causal pathways exist through which diet can influence brain functioning. For instance, intake of healthy foods is known to decrease inflammation markers, and thereby decrease neuroinflammation (Wang et al., 2016, Johnson et al., 2020). Similarly, HPA-axis activation is shown to be downregulated though decreased proinflammatory cytokines and glucocorticoid levels in response to dietary alterations. Upregulated histone acetylation and BDNF expressions can also lead to epigenetic changes in response to dietary interventions (Johnson et al., 2020). ). In these ways, we postulate that unhealthy dietary composition over the lifespan could lead to altered brain structure and functioning, which may subsequently lead to issues like increased disinhibitory behavior. Further studies in the UKBiobank and other samples should take these pathways into account to further complete the causal relations between diet, disinhibition and the brain.”
